

# Temporal overlaps of feral cats with prey and competitors in primary and human-altered habitats on Bohol Island, Philippines

Vlastimil Bogdan, Tomáš Jůnek and Pavla Jůnková Vymyslická

Department of Ecology/Faculty of Environmental Sciences, Czech University of Life Sciences Prague, Prague, Czech Republic

## ABSTRACT

The vertebrate fauna of the Philippines, known for its diversity and high proportion of endemic species, comprises mainly small- to medium-sized forms with a few large exceptions. As with other tropical ecosystems, the major threats to wildlife are habitat loss, hunting and invasive species, of which the feral cat (*Felis catus*) is considered the most damaging. Our camera-trapping study focused on a terrestrial vertebrate species inventory on Bohol Island and tempo-spatial co-occurrences of feral cats with their prey and competitors. The survey took place in the Rajah Sikatuna Protected Landscape, and we examined the primary rainforest, its border with agricultural land, and rural areas in the vicinity of villages. Altogether, over 2,885 trap days we captured 30 species of vertebrates–10 mammals (including *Sus philippensis*), 19 birds and one reptile, *Varanus cumingi*. We trapped 81.8% of expected vertebrates. Based on the number of events, the most frequent native species was the barred rail (*Gallirallus torquatus*). The highest overlap in diel activity between cats and potential prey was recorded with rodents in rural areas ($\Delta = 0.62$); the lowest was in the same habitat with ground-dwelling birds ($\Delta = 0.40$). Cat activity was not recorded inside the rainforest; in other habitats their diel activity pattern differed. The cats' activity declined in daylight in the proximity of humans, while it peaked at the transition zone between rainforest and fields. Both rodents and ground-dwelling birds exhibited a shift in activity levels between sites where cats were present or absent. Rodents tend to become active by day in cat-free habitats. No cats' temporal response to co-occurrences of civets (*Paradoxurus hermaphroditus* and *Viverra tangalunga*) was found but cats in diel activity avoided domestic dogs (*Canis lupus familiaris*). Our first insight into the ecology of this invasive predator in the Philippines revealed an avoidance of homogeneous primary rainforest and a tendency to forage close to human settlements in heterogeneous habitats. A detailed further investigation of the composition of the cat's diet, as well as ranging pattern, is still needed.

Corresponding author
Tomáš Jůnek, tom_junek@yahoo.com

## INTRODUCTION

The Philippine Archipelago is considered a global biodiversity hotspot, known for its high proportion of endemic species (*Ambal et al., 2012*). The terrestrial vertebrate taxa, which primarily encompass small to medium sized species, inhabit more than 7,100 islands. These species include at least 213 mammals (*Heaney et al., 2010*), 674 birds (*Lapage, 2015*), 270 reptiles and 111 amphibians (*BREO, 2015*).

Similar to other oceanic islands, the predominantly small fauna of the Philippines suffers from the presence of competing invasive species, such as *Rattus* spp., and the feral cat (*Felis catus*). The cat is listed as the most widespread and probably most damaging of the four carnivores included on the list of the 100 worst invasive species (*Lowe et al., 2000*). At least 175 vertebrates are threatened or have been driven to extinction by feral cats on at least 120 islands (*Medina et al., 2011*). Meta-analysis has revealed that the negative impact of feral cats is largest for insular endemic mammals, and is exacerbated by the presence of invasive cat prey species such as mice (*Mus musculus*) or rabbits (*Oryctolagus cunuculus*) (*Nogales et al., 2013*). The cat is widely kept as a pet by people throughout the Philippines and can be found foraging in every habitat (*Duffy & Capece, 2012*). Despite the general prevalence of cats in the Philippine landscape, there is a noticeable lack of knowledge regarding the cat's impact on the biodiversity of this archipelago.

Cats feed on a wide range of animals, from arthropods, reptiles and birds to mammals the size of a rabbit (*Pearre, Maass & Maass, 1998*). In Australia alone, with a variety of animals of similar size such as those found in the Philippines, 400 prey species consumed by cats have been recorded (*Doherty et al., 2015*). In the Philippines, members of the orders Chiroptera and Rodentia are the most numerous mammalian species (*Heaney et al., 2010*). A wide range of terrestrial and arboreal rodents with body mass ranging from the 15-g *Musseromys* spp. to the 2.6-kg *Phloeomys* spp. risk predation by cats. Only adult individuals of *Phloeomys* and *Hystrix pumila* (*Heaney et al., 2010*) exceed the potential prey dimensions. According to size and niche, members of the Tupaiidea (treeshrews), Erinaceidae (moonrats) and Soricidae family (shrews) should be listed as mammalian prey for cats. Similarly, the smallest Philippine primate, *Tarsius syrichta*, which inhabits Bohol and other islands of the Mindanao faunal region, can be included (*MacKinnon & MacKinnon, 1980*).

On Bohol Island (3,269 km$^2$), as on the other Philippine islands, bats and rodents dominate among local mammals. The small mammalian fauna consists of one insectivorous species and nine species of rodents, including the introduced *Mus musculus, Rattus rattus, Rattus norvegicus, Rattus tanezumi* and *Rattus exulans* (*Heaney et al., 2010*). The avifauna of Bohol numbers 235 species, with Passeriformes forming the largest sub-group at 83 species. Bohol is also home to 14 ground-dwelling bird species inhabiting the woody or bushy inland habitats potentially affected by cats (*Kennedy, 2000*).

Along with dogs (*Canis lupus familiaris*), possible competitors of cats on Bohol include two mammalian carnivores, Asian palm civet (*Paradoxurus hermaphroditus*) and Malayan civet (*Viverra tangalunga*) (*Heaney et al., 2010*) and two reptile species: yellow-headed water monitor (*Varanus cumingi*) and reticulated python (*Python reticulatus*) (*BREO, 2015*). To our knowledge, no predation between cat and civets has been published.

 

The timing of activity of mammalian predators is a well discussed topic (e.g., *Palomares & Caro, 1999*; *Tambling et al., 2015*). Time-stamped records from camera traps allow for detailed insights into the time budget and temporal coexistence of animals across trophic guilds, seasons, etc. (*Rowcliffe et al., 2014*), and recent camera trapping studies have successfully examined overlaps in diel activity patterns (*Ridout & Linkie, 2009*), confirming significant activity overlap between carnivores and their preferred prey (*Harmsen et al., 2009*; *Lucherini et al., 2009*; *Sweitzer & Furnas, 2016*) and suggesting predator behavior to reduce foraging energy expenditure (*Foster et al., 2013*). In their role as mesopredators cats must optimize their use of time not only to encounter prey but also to cope with a sympatric superior predator (*Brook, Johnson & Ritchie, 2012*). The combination of partitioning of habitat, prey size and a 24-h daily cycle is thought to be a complex mechanism allowing competing felids to coexist in different animal communities (*Di Bitetti et al., 2010*; *Foster et al., 2013*; *Silmi, Anggara & Dahlen, 2013*; *Sunarto et al., 2015*). For example, low overlap in activities has been found between marbled cat (*Pardofelis marmorata*) and clouded leopard (*Neofelis nebulosa*) in Thailand (*Lynam et al., 2013*). *Wang & Fisher (2012)* also confirmed higher segregation of diel activities of cats with respect to dingoes during wet months. The particularly suppressive effect of an apex carnivore on invasive populations of cats is considered an important conservation issue (*Brook, Johnson & Ritchie, 2012*; *Lazenby & Dickman, 2013*; *Doherty, Bengsen & Davis, 2015*).

We conducted a camera-trap survey on Bohol Island in an attempt to uncover tempo-spatial co-occurrences of terrestrial vertebrate species on regularly used trails and to confirm the presence of cats in the protected primary rainforest (Zone I), a transition zone along the border of the primary rainforest with the agricultural landscape (Zone II), and inside the rural landscape in the proximity of human settlements (Zone III). Our objectives were to: (1) create a general inventory of camera-trapped taxa; (2) model the species accumulation curve using previous knowledge of the possible number of mammalian, avian and reptile species detectable by camera-traps; and (3) compare the diel activity levels of cats with those of potential prey and competitors.

## MATERIALS & METHODS

### Study site

Our study was conducted under research permit No. 2014-04, issued by DENR, Region VII, Philippines, between July 2nd and December 4th, 2014 in the surroundings of the town of Bilar, Bohol Island, Philippines. The landscape consists of a mixture of distinctive flat rural areas near human settlements, used as rice fields and plantations for various crops, steep karst hills covered by brush and secondary forest, and primary rainforest in protected areas. The town of Bilar lies between two conservation areas, the Rajah Sikatuna Protected Landscape (RSPL) and the Loboc River Watershed Forest Reserve. RSPL is the second largest protected sanctuary on Bohol, covering 11,034 ha of a mostly hilly limestone environment rich in characteristic landforms such as ravines, sinkholes and caves. The altitude in RSPL varies between 300 and 826 m above sea level. The forest canopy is multi-layered, with trees reaching up to 20 m in height. Members of the families Dipterocarpaceae, Moraceae

**Table 1** Summary of the camera trap deployment in the area of the Rajah Sikatuna Protected Landscape, Bohol, Philippines.

| Zone | Site | Date | n camera traps | n trap-days | Range of distances between traps (m) |
|------|------|------|----------------|-------------|--------------------------------------|
| I | SP | 12.7.–30.7.2014 | 12 | 204 | 60–514 |
| I | SF | 30.7.–14.8.2014 | 12 | 173 | 38–307 |
| I | WS | 1.8.–15.8.2014 | 9 | 125 | 37–265 |
| II | BI | 2.8.–4.12.2014 | 10 | 850 | 28–395 |
| II | LS | 12.7.–4.12.2014 | 7 | 536 | 48–174 |
| II | BU | 2.8.–4.12.2014 | 4 | 383 | 38–44 |
| III | HB | 5.7.–31.7.2014 | 10 | 224 | 25–236 |
| III | SU | 2.7.–31.7.2014 | 16 | 390 | 23–139 |

and Melicacea dominate the canopy. Certain regions of RSPL have been reforested with white teak (*Gmelina arborea*) and Honduras mahogany (*Swietenia macrophylla*) (*Barcelona et al., 2006*). The average annual precipitation reaches 1,600 mm; the rainy season typically lasts from June to December, with an increase in precipitation to 200 mm per month. The driest month is April when approximately 40 mm of rain falls.

## Sampling design

We monitored three types of landscape typical of tropical regions and deployed cameras in groups, one camera per location, at eight trapping sites (Fig. 1): Zone I—protected primary rainforest including the Watershed Forest Reserve (site WS), interior of RSPL (site SP) and abandoned farms in the early stages of succession into RSPL (site SF); Zone II—transition zone between the primary rainforest of RSPL and rice fields close to the village of Bulak (site BU), transition zone between RSPL and rice and corn fields close to Logarita Springs (site LS), and transition zone between RSPL and the farms of the village Binantay (site BI); Zone III—mixture of brush and degraded forest and plantations on the edge of the village of Subayon (site SU), and at Bohol Habitat Conservation Center on the edge of the town of Bilar (site HB). Details on camera traps' deployment and duration of sampling are shown in Table 1.

## Sampling procedures

We used 41 weatherproof infrared digital camera traps –29 units of Ltl Acorn 5210MC (Shenzhen Ltl Acorn Electronics Co., Ltd.) and 12 units of SPYPOINT IR7 (SPYPOINT[MD], G.G. Telecom). Prior to the study, we tested both types of cameras in a week-long trial which was focused on the difference in detection rates for moving objects. No difference larger than 10 % between numbers of independence events was found. Both types of cameras were also used in every habitat to avoid a bias from site-specific detection rates. Cameras were set up to perform the same delay between recordings –SPYPOINT to take two images with a delay of 10 s between consecutive triggering, and Ltl Acorns to take one picture followed by a 5 s video, with a 5 s delay between triggering. Video sequences served as an additional tool for the identification of species.

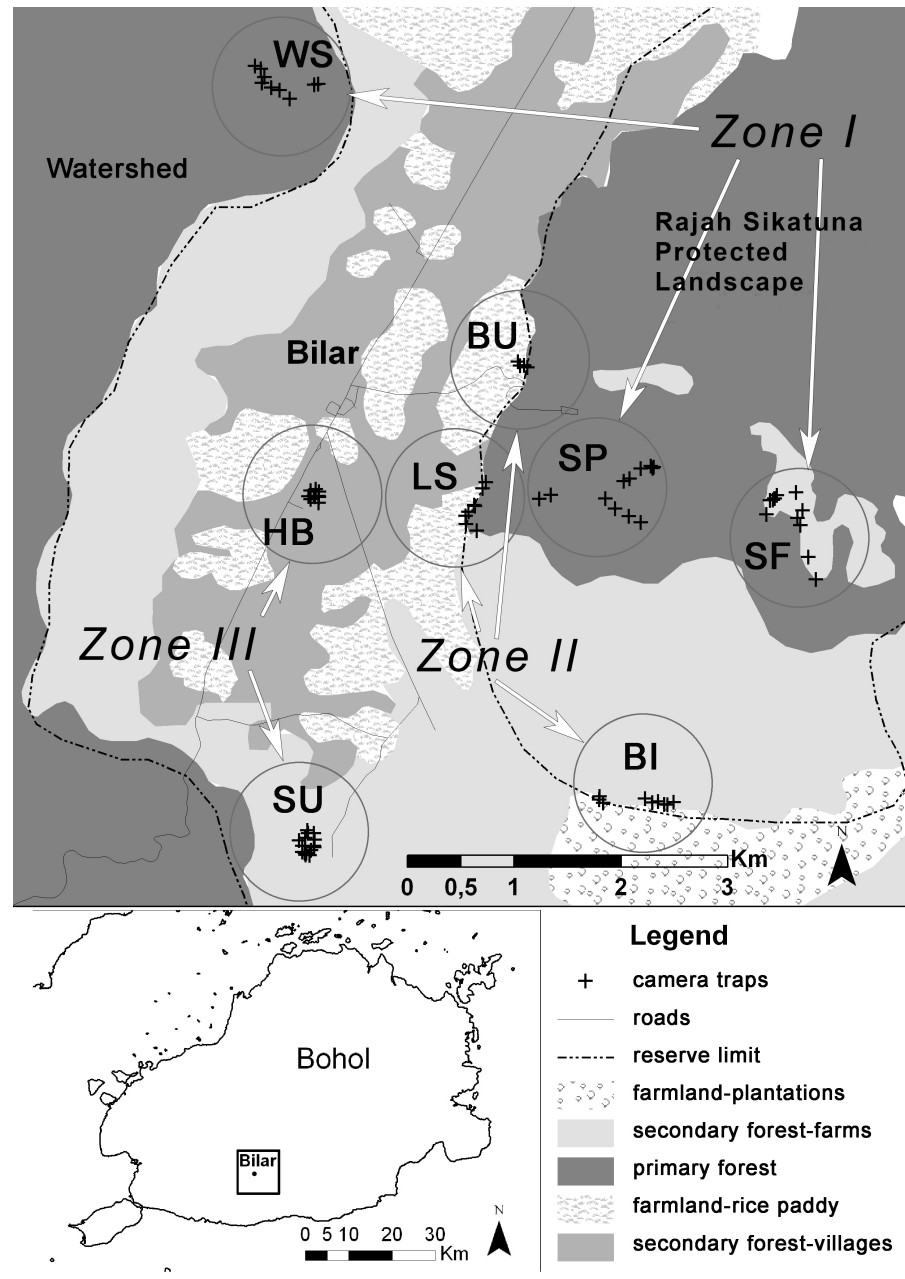

**Figure 1  Schematized map of the study area in the municipality of Bilar, Bohol, Philippines.** Circles highlight eight sites surveyed from July to December 2014.

We placed all cameras opportunistically on the most frequented trails or their junctions and, according to the expected size of target vertebrates, we fastened cameras with a belt onto the trunks of trees or bushes nearest to the trail, at a height of up to 0.5 m, with a focal point approximately 2 m from the lens. All cameras were active 24 h a day; all records in infrared mode were available only in a black-and-white version. No bait was used.

## Identification of taxa

Two observers, VB and TJ, independently identified all species visually from images and videos; the results were mutually crosschecked, and disagreeing or unidentifiable records were excluded from the analysis. Based on available databases (*Heaney et al., 2010*; *BREO, 2015*; *Lapage, 2015*), we made a list of terrestrial mammalian and avian ground-dwelling species known or expected to occur on Bohol (Table 2). From reptiles, we included only the largest four-legged taxon, the yellow-headed water monitor (*Varanus cumingi*). Members of the order Chiroptera and the strictly arboreal Philippine colugo (*Cynocephalus volans*) were *a priori* omitted. The conservation status of each species was assessed following *IUCN (2015)*.

Due to the limited nature of the recordings, for the identification process and the calculation of a species accumulation curve all taxa the size of a mouse (*Mus musculus* and also the insectivorous *Crocidura beatus*) were pooled into the group called 'mice', and all species of rats (*Rattus* spp. and *Bullimus bagobus*) into the group 'rats'. In addition, both known species of squirrels (*Exilisciurus concinnus* and *Sundasciurus philippinensis*) were grouped into one taxon: 'squirrels'. In total, the list consisted of eight taxa of mammals, 13 birds and one reptile. For purposes of *overlap* analyses between cats and their competitors and prey, we pooled both native carnivore species into a group called 'civets' and put mice, rats and squirrels into the group 'rodents'. Ground-dwelling species of birds were the second analyzed group of prey; dogs were accordingly examined as competitors.

## Data analysis

Photographs were defined as events (or activity records) when the delay between two consecutive images of an individual exceeded 10 min. The same individual could theoretically trigger more than one camera within 10 min. For each species and Zone, in Table 2 we reported occurrences of species at cameras represented by events (*Lazenby & Dickman, 2013*).

We used a species accumulation curve based on the cumulative number of camera-trapping days, computed in EstimateS Version 9.1.0 (*Colwell, 2013*), to find out if our survey lasted a sufficient number of days to capture the 22 expected terrestrial vertebrate species (including three pooled groups) known from Bohol. We followed *Tobler et al. (2008)* and calculated well-performing estimators of species richness: the non-parametric abundance-based estimator ACE, and the non-parametric incidence-based estimators ICE and Jackknife 1. An abundance-based rarefaction approach with 95% confidence intervals and 1,000 random iterations of sample order was used.

The pair-wise temporal overlap of selected activity patterns was analyzed using the R statistical environment package '*overlap*' (*Meredith & Ridout, 2014*). Following *Ridout & Linkie (2009)*, we applied kernel density estimation on circular data pooled within all study sites. Density of activity ($y$-axis) uses a von Mises kernel, corresponding to a circular distribution, and is based on recorded time of each event on 24-h $x$-axis. The coefficient of overlap ($\Delta$) was calculated with a smoothing parameter of 1.0. We used a smoothed bootstrap of 10,000 resamples to determine standard errors and 95% confidence intervals. We only analyzed combinations of pairs of species, which scored at least 30 events in the activity record (MS Ridout, pers. comm., 2015) in a given environment. The number

**Table 2   List of species recorded during a survey in the area of the Rajah Sikatuna Protected Landscape, Bohol, Philippines.** Values represent number of events of species recorded in each zone.

| Common name | Scientific name | Zone I | Zone II | Zone III | Site |
|---|---|---|---|---|---|
| Philippine warty pig[a] | *Sus philippensis* | 1 | 0 | 0 | SF |
| Common palm civet[a] | *Paradoxurus hermaphroditus* | 10 | 43 | 6 | Su,Bi,LS,Bu,SP,SF,Ws |
| Malay civet[a] | *Viverra tangalunga* | 1 | 8 | 7 | Ha,Bi,LS,SP |
| Long-tailed macaque[a] | *Macaca fascicularis* | 3 | 4 | 0 | LS,Bu,Ws |
| Philippine tarsier[a] | *Tarsius syrichta* | 0 | 1 | 2 | Su,LS |
| Philippine tree squirrel[a] | *Sundasciurus philippinensis* | 4 | 42 | 1 | Su,LS,Bu,SP,SF,Ws |
| Dog | *Canis lupus familiaris* | 14 | 39 | 91 | Su,Ha,Bi,LS,Bu,SP |
| Cat | *Felis catus* | 0 | 97 | 83 | Su,Ha,Bi,LS |
| Rat[a] | *Rattus* spp. | 32 | 242 | 47 | Su,Ha,Bi,LS,Bu,SP,SF,Ws |
| Mice[a] | *Mus* spp. | 5 | 217 | 5 | Su,Ha,Bi,LS,Bu,SP,SF,Ws |
| Yellow-headed water monitor[a] | *Varanus cumingi* | 1 | 6 | 5 | Su,LS,SF |
| Hooded pitta[a] | *Pitta sordida* | 0 | 0 | 41 | Ha |
| Red-bellied pitta[a] | *Pitta erythrogaster* | 0 | 42 | 0 | Bi,LS |
| Azure-breasted pitta[a] | Pitta steerii | 1 | 0 | 0 | Ws |
| Striated wren-babbler[a] | *Ptilocichla mindanensis* | 1 | 9 | 0 | LS,Ws |
| Red junglefowl[a] | *Gallus gallus* | 0 | 5 | 14 | Su,LS,Bu |
| Barred rail[a] | *Gallirallus torquatus* | 0 | 4 | 179 | Su,Ha,Bi,LS,Bu |
| Slaty-legged crake[a] | *Rallina eurizonoides* | 0 | 0 | 15 | Su,Ha |
| Ruddy-breasted crake[a] | *Zapornia fusca* | 0 | 2 | 1 | Su,Bu |
| Plain bush-hen[a] | *Amaurornis olivacea* | 0 | 0 | 10 | Su |
| Black-faced coucal | *Centropus melanops* | 2 | 4 | 2 | Su,Bi,LS,Bu,SF,Ws |
| Philippine coucal | *Centropus viridis* | 0 | 2 | 3 | Su,Bu |
| Emerald dove | *Chalcophaps indica* | 7 | 7 | 16 | Su,Ha,Bi,LS,Bu,SF,Ws |
| Philippine magpie-robin | *Copsychus mindanensis* | 0 | 3 | 1 | Ha,Bi |
| Mindanao bleeding-heart | *Gallicolumba crinigera* | 1 | 2 | 0 | LS,SP |
| Hair-crested drongo | *Dicrurus hottentottus* | 1 | 0 | 0 | SF |
| Besra | *Accipiter virgatus* | 0 | 1 | 1 | Su,Bi |
| Philippine hawk-owl | *Ninox philippensis* | 0 | 0 | 1 | Ha |
| Yellow-breasted tailorbird | *Orthotomus samarensis* | 0 | 1 | 0 | Bi |
| Domestic chicken | *Gallus gallus domeaticus* | 10 | 254 | 9 | Su,Bi,LS,Bu,SP,SF |

**Notes.**
[a]Species those expected for the species accumulation curve.

of events used for calculation of the activity pattern overlap for each analyzed group of animals and each location is shown in Table 3.

## RESULTS

### Species inventory

During the whole survey period, lasting 155 days, we accumulated 2,885 trap days and 2,034 events. The combined capture rate across all sites was 73.1 events per 100 trap days. The list of all 30 animal taxa recorded is shown in Table 2.

**Table 3  Number of events used for calculation of activity pattern overlap for each analyzed group of animals and each location in the area of the Rajah Sikatuna Protected Landscape, Bohol, Philippines.** Values in parentheses show the zone-specific relative abundance index (events/total trap days in zone*100). Dashes denote unprocessed entries.

|  | Zone I | Zone II | Zone III | All sites | No cats | With cats |
|---|---|---|---|---|---|---|
| Cats | 0 (0) | 67 (4.57) | 83 (13.52) | 150 (5.81) | – | – |
| Dogs | 14 (2.79) | 37 (2.53) | 90 (14.66) | 141 (5.46) | – | – |
| Rodents | 41 (8.17) | 480 (32.76) | 47 (7.65) | 568 (22.00) | 41 (8.18) | 527 (25.35) |
| Ground-dwelling birds | 12 (2.39) | 242 (16.52) | 263 (42.83) | 517 (20.03) | 12 (2.39) | 505 (24.29) |
| Civets | 11 (2.19) | 16 (1.09) | 13 (2.12) | 40 (1.39) | – | – |

The most frequent native species was the barred rail (*Gallirallus torquatus*), captured in 183 independence events. We did not record four expected bird species: *Megapodius cumingii*, *Coturnix chinensis*, *Turnix sylvaticus* and *Gallinago megala*. On the other hand, we confirmed the survival of the Philippine warty pig (*Sus philippensis*). Given its size, it was probably a male individual that was captured, only once, on three images on August 9th (6:35 pm) in a mud wallow in the interior of RSPL.

We found that feral cats most often occurred in the Zone II and III, and were absent inside the primary forest. A similar trend was found for ground-dwelling birds. Most rats and other small mammals were recorded in the transition Zone II between the RSPL forest and agricultural land. Along with feral cats and domestic dogs, we also recorded all three medium-sized mammals occurring on Bohol—the common palm civet (59 events), Malay civet (16 events) and long-tailed macaque (*Macaca fascicularis*) (7 events). Humans were also captured but excluded from the analysis.

Within all eight sampling sites, we captured 18 of 22 expected target taxa, which corresponds to a success rate of 81.8% of the species inventory (100% of mammals and reptiles, 69.2% of birds). We used these 18 taxa for calculating the species accumulation curve (Fig. 2). The mean estimated species richness computed in EstimateS was 19.7 species (ACE = 19.6, ICE = 19.5 and Jackknife 1 = 20.0). We recorded 15.89 species (72.2% of expected species) in 1,000 trap days. The eight target species of mammals were captured in 1,723 trap days; similarly, nine ground-dwelling birds were recorded within 1,435 trap days.

## Temporal overlaps

We recorded cats only in transition Zone II and in the rural landscape close to human settlements (Zone III). Diel activity patterns of cats differed among zones (Fig. 3). Cats showed a decrease in late-afternoon activity near villages, whereas activity in the transition area peaked right before noon. Generally, the activity of cats by daylight was higher in transition zones; in Zone III cats were recorded mainly at night.

The highest overlap in activity patterns between cats and rodents (Table 4) was found in the rural landscape of Zone III, and between cats and ground-dwelling birds in transition Zone II (Fig. 4).

Both categories of potential prey showed shifts in temporal occurrence within sites, based on the presence of cat (Fig. 5). As seen, the peaks of rodent activity decreased in the

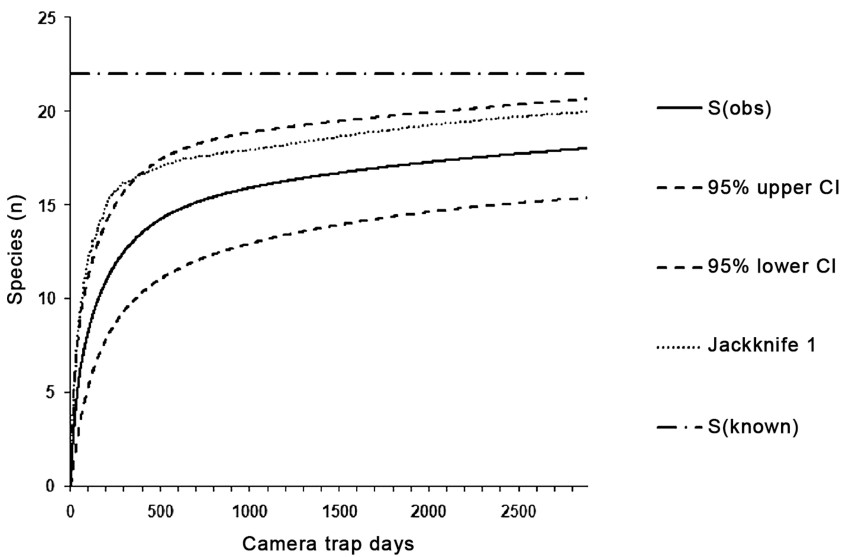

**Figure 2** **The species accumulation curves with 95% CIs for species captured in all categories of environment in 2,885 trap days.** The dashed-and-dotted line marks the known number of species, while the dotted lines represent species richness estimated by Jackknife 1 in EstimateS (*Colwell, 2013*).

## cats zone II vs. zone III

**Figure 3** **Overlap between diel activity patterns of cats in transition Zone II (dashed line) and rural Zone III.** The number represents the coefficient of overlap ($\Delta$), with standard error in parentheses.

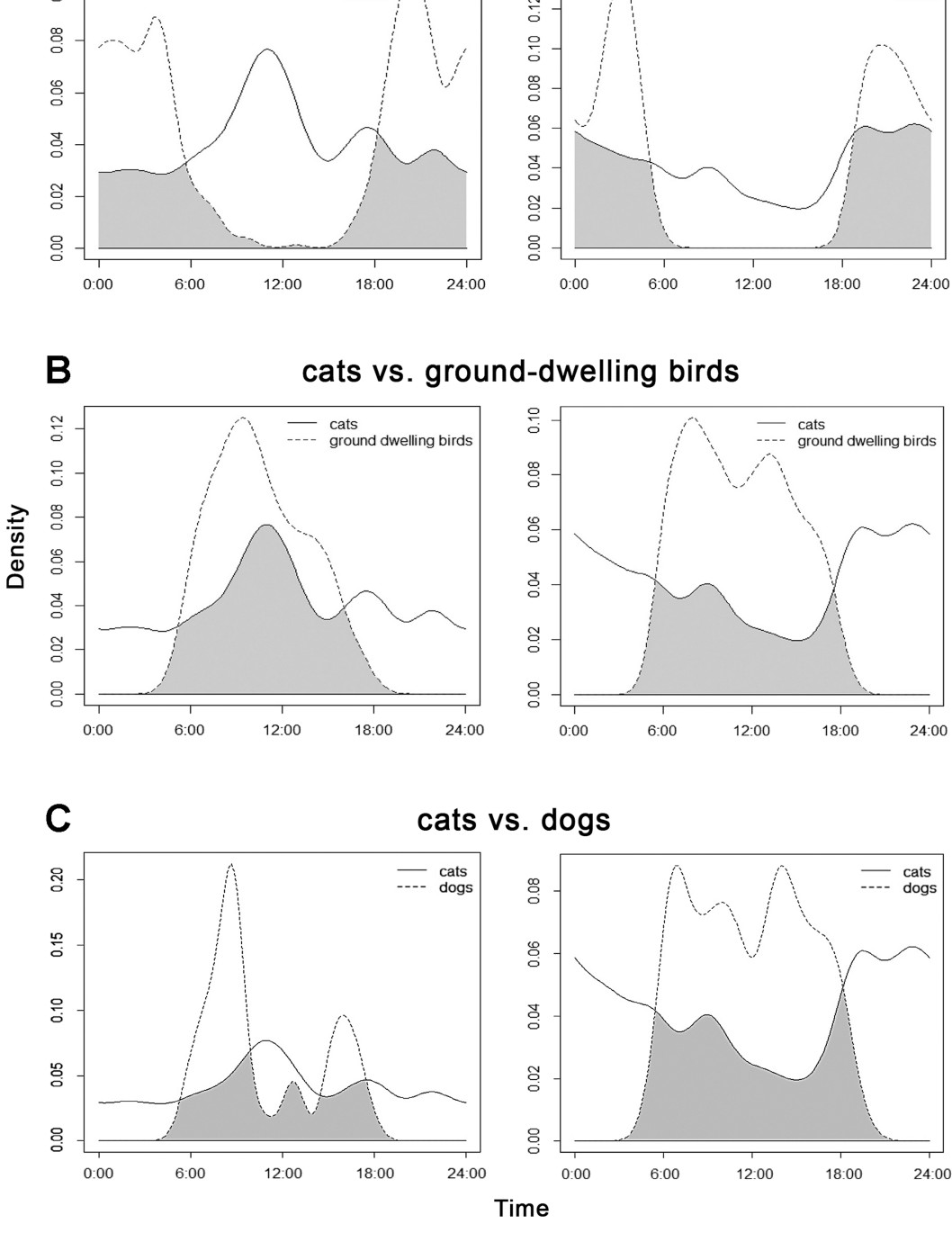

**Figure 4** Overlap between the diel activity patterns of cats with (A) rodents, (B) ground-dwelling birds and (C) dogs in transition Zone II and rural Zone III.

**Table 4** Activity pattern overlaps between cats, their potential prey (rodents and ground-dwelling birds) and competitors (dogs and civets) in transition Zone II, rural Zone III and among all sites surveyed in the area of the Rajah Sikatuna Protected Landscape, Bohol, Philippines.

|  | Site | Overlap Δ | *SE* | 95% l*CI* | 95% u*CI* |
|---|---|---|---|---|---|
| Cats vs. rodents | Zone II | 0.48 | 0.023 | 0.37 | 0.58 |
|  | Zone III | 0.62 | 0.002 | 0.52 | 0.73 |
| Cats vs. ground-dwelling birds | Zone II | 0.61 | 0.019 | 0.50 | 0.71 |
|  | Zone III | 0.40 | 0.041 | 0.30 | 0.50 |
| Cats vs. dogs | Zone II | 0.50 | 0.052 | 0.36 | 0.62 |
|  | Zone III | 0.45 | 0.041 | 0.35 | 0.56 |
| Cats vs. civets | All sites | 0.55 | 0.067 | 0.45 | 0.64 |

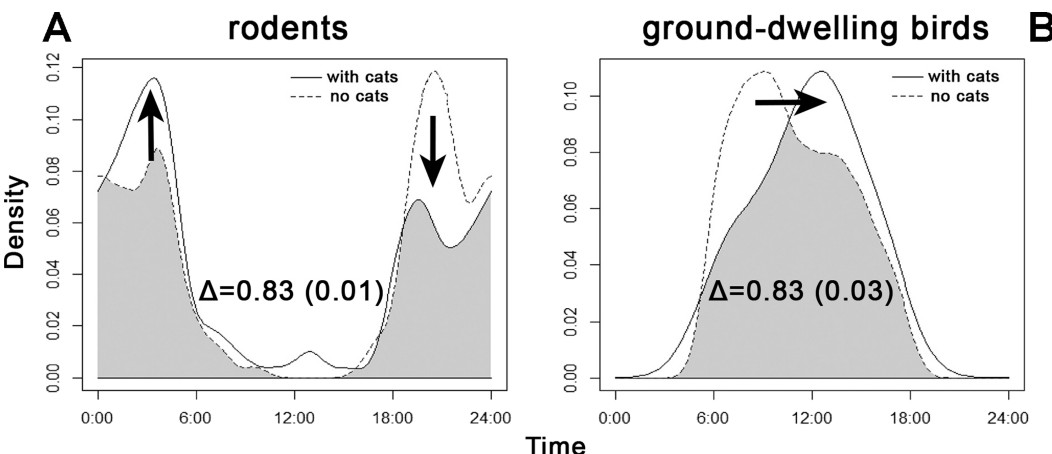

**Figure 5** Overlap between the diel activity patterns of (A) rodents and (B) ground-dwelling birds at sites with and without the presence of cats. The numbers represent coefficients of overlap (Δ) with standard errors in parentheses. Arrows indicate shifts in activity levels if cats are present.

hours before sunrise and increased after sunset, whereas the activity of ground-dwelling birds peaked about 4 h sooner at sites where cats were not recorded.

Cats showed the second lowest overlap among all groups with dogs in Zone III (Table 4) where dogs were dominant and active during the day. In Zone II these two animals appeared to peak in their activity at different times: dogs were most active in the morning and late afternoon, whereas cats peaked before noon (Fig. 4).

The overlap between the diel activity patterns of cats and both species of civets is shown in Fig. 6. Cats exhibited roughly consistent activity throughout a 24-h period, with no apparent shift caused by the nocturnal occurrence of sympatric civets.

## DISCUSSION

According to our knowledge, to date no study of the behavior and ecology of feral cats has been conducted in the Philippines, nor any camera-trap-based species inventory on Bohol. With the exception of the Philippine pygmy squirrel, *Exilisciurus concinnus*, we were able to

capture and identify every non-volant mammalian species recorded as occurring on Bohol larger than a mouse, including an individual of *Sus philippensis*, which is considered to be close to extinction (*Oliver, 1993*), even by local people. Camera traps captured 81.8% of known ground-dwelling mammalian, avian, and reptilian species, similar to the 86 % captured in the Amazon rain forest (*Tobler et al., 2008*) or 89% in the lowland rainforest of Borneo (*Bernard et al., 2013*); both those camera-trapping studies were restricted to mammals. In addition, the initially steep shape of our general species accumulation curve corresponds with studies conducted in tropical ecosystems and confirms the robustness of the approach. Similarly to *Rovero et al. (2014)*, we captured the majority of selected species in 1,000 trap days, considered a reliable threshold enabling the detection of rare species (*O'Brien, 2011*).

The absence of cats in the interior of primary rainforest seems not to be driven by distance from the nearest human settlements, given that all three monitored sites were up to approximately 3 km from houses. We suggest that the absence of preferred features and habitats in the rain forest may have resulted in camera traps failing to capture cats. Cats typically use a mixture of vegetation cover at ground level which provides both cover and open space for observing their prey; such habitat may increase hunting success (*Doherty, Bengsen & Davis, 2015*). The habitat heterogeneity hypothesis by *Tews et al. (2004)* predicts that heterogeneous habitats offer a greater diversity and density of potential prey than homogeneous ones, which could be conceivable for cats. Linear features in space (e.g., tree lines, roads and other corridors) are generally considered to maximize cat's detectability (*Crooks, 2002*; *Bengsen, Butler & Masters, 2012*). We would expect to record cats in primary forest mostly on trails (*Trolle & Kéry, 2005*; *Harmsen et al., 2010*; *Anile et al., 2014*) but they could disperse into the undergrowth on paths that are undetected.

The presence of competing, potentially dangerous predators in primary forest is unlikely to explain the absence of cats. Dogs and both species of civets were equally present in all three zones. The common palm civet and Malay civet are omnivorous with a distinctive nocturnal activity pattern (*Jennings et al., 2009*) but they forage in the habitat of cats, and given their size we consider them to be competitors of cats. Nonetheless, cats do not show any temporal avoidance, indicating no interspecies competition, which has evolved during almost a 500-year co-existence (*Jubair, 1999*). For a more comprehensive view of possible niche partitioning, as found for example between felids on Sumatra (*Sunarto et al., 2015*), a camera-trapping study should be conducted on Negros, where the Visayan leopard cat (*Prionailurus bengalensis* ssp. *rabori*) occurs as a regional direct competitor (*IUCN, 2015*).

Our results (Tables 2 and 3) show that species richness and availability of both prey categories (rodents and birds) was higher, nearly by orders of magnitude, in both human-altered zones than in primary rainforest. We attribute this to the variety of vertebrate and invertebrate prey, which is more abundant in heterogeneous landscapes. In addition, as suggested by *Lozano et al. (2003)*, feral cats use a wide range of habitat components to meet their different activity requirements (e.g., hunting, resting), and this landscape offers a mixture of agricultural features with secondary growth, infrastructure and potential human subsidies (*Ferreira et al., 2011*). Our data clearly support such a tendency to forage relatively close to human settlements, however we were not able to determine from our records whether a photographed animal was feral or domestic.

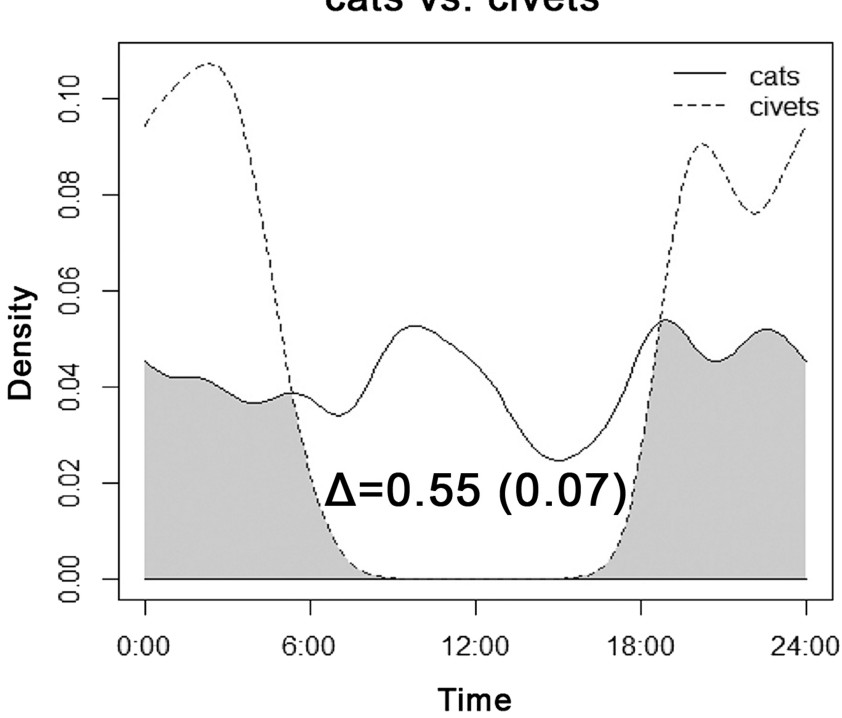

**Figure 6 Overlap between the diel activity patterns of cats and both species of civets at all sites of their occurrence.**

Although the diel activity pattern of cats was roughly consistent over 24-h periods (Fig. 6), a detailed analysis of zones revealed a decline in diurnal activity in the vicinity of villages, in contrast to an apparent activity peak before noon in Zone II (Fig. 3). Both Zones II and III offered more-or-less the same number of prey species (Table 2). Compared with Zone II, we hypothesize that more uniform diurnal activity of relatively abundant dogs dissuaded cats from daytime foraging in Zone III (Fig. 4). Also subsidies provided by humans in villages could influence cats to remain inactive in shelters and forage at night. We did not detect any sign of cats being spatially excluded by dogs, but our results support findings that cats optimize their timing of hunting behavior to when dogs are less active, hence avoiding potentially dangerous encounters (*Brook, Johnson & Ritchie, 2012*; *Wang & Fisher, 2012*). Cats were more diurnal in Zone II. This could be explained by the same factor, because the zone-specific relative abundance index of dogs was two times lower than the index of cats than in Zone III (Table 3). So cats could respond both to lower disturbance from dogs and to higher diurnal availability of rodents in Zone II (Fig. 4). Other prey not detected by cameras such as insects or lizards might also be present (*Bonnaud et al., 2011*).

Prey species showed shifts in diel activity patterns between sites where cats were, or were not, present (Fig. 5). When cats were absent, rodents tended to forage visibly by day, while the activity of ground-dwelling birds peaked about 4 h later. It is difficult to interpret the shift in bird activity; data from sites without cats were considered too scarce to perform a reliable analysis. Rodents shift their activities to become nocturnal if cats are present and

more diurnal (*Doherty et al., 2015*). This raised the question of whether almost 500 years of cat presence in the Philippines has driven adaptive mechanisms of prey and competitors to cope with a new predator or not. Our results suggest that this already happened, similar to the 4000-year history of the dingo in Australia (*Carthey & Banks, 2012*). Nevertheless, we believe that further research is needed, especially throughout all seasons.

Knowledge of feral cat diet is paradoxically the least researched in tropical habitats with the richest terrestrial biodiversity (*Doherty et al., 2015*; *Doherty, Bengsen & Davis, 2015*). Our findings reveal the first tempo-spatial co-occurrences between feral cats and their potential prey in a typical mixture of Philippine landscapes. We suggest feral cats' temporal avoidance of dogs as the apex predator. We confirm that camera traps are capable of capturing small-bodied fauna, ground-dwelling birds and highly elusive species, such as *Sus philippensis*, as well. Endangered Philippine fauna exposed to invasive species should rapidly become the target of a broad and long-term camera-trapping inventory survey. For an in-depth knowledge of the dietary intake of feral cats in the Philippines, DNA analysis of scat is recommended as a priority for researchers (*Nogales et al., 2013*). In addition, collared and GPS-tracked cats would provide information about habitat use and the size of home ranges. Finally, attention should be paid to the cultural value of cats kept as pets within Philippine society, to inform eradication strategies.

## ACKNOWLEDGEMENTS

We would like to acknowledge the following persons and organizations which made our study possible: Mr. Isabelo R. Montejo, regional director, and Mr. Eusalem S. Quiwag from the Department of Environment and Natural Resources (DENR), Region VII, Philippines; the staff of Rajah Sikatuna PL; Ms. Cristy Burlace and staff from the Habitat Bohol; Dr. Petr Anděl from the Czech University of Life Sciences Prague; Ms. Monika Drimlová for tireless field work; and two anonymous reviewers for comments on this manuscript.

### Funding

The project was financially supported by grants GA FZP, reg. No. 20144257, and GA FZP, reg. No. 20134247, provided by the Czech University of Life Sciences Prague. The funders had no role in study design, data collection and analysis, decision to publish, or preparation of the manuscript.

### Grant Disclosures

The following grant information was disclosed by the authors:
GA FZP: 20144257, 20134247.

### Competing Interests

The authors declare there are no competing interests.

## Author Contributions

- Vlastimil Bogdan and Tomáš Jůnek conceived and designed the experiments, performed the experiments, analyzed the data, contributed reagents/materials/analysis tools, wrote the paper, prepared figures and/or tables, reviewed drafts of the paper.
- Pavla Jůnková Vymyslická conceived and designed the experiments, performed the experiments, contributed reagents/materials/analysis tools, reviewed drafts of the paper.

## Field Study Permissions

The following information was supplied relating to field study approvals (i.e., approving body and any reference numbers):

Department of Environment and Natural Resources (DENR), Region VII, Philippines; Research permit No. 2014-04

## Data Availability

Jůnek, Tomáš (2016): Dataset from Bohol. Figshare. https://dx.doi.org/10.6084/m9.figshare.2245810.

## Supplemental Information

Supplemental information for this article can be found online at http://dx.doi.org/10.7717/peerj.2288#supplemental-information.

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
