# Peer review of "Temporal overlaps of feral cats with prey and competitors in primary and human-altered habitats on Bohol Island, Philippines"

_PeerJ, doi:10.7717/peerj.2288_

## Round 0.1 · original submission · Major Revisions

Dear Dr Junek,

I've received reports from two reviewers and will paste my own comments below. We all feel your study is of interest and has merit, but needs some revisions before it can be published. In general, you need to align your objectives better with your content, and in the Introduction focus more on why camera traps were used and how the information from camera traps tells us about activity levels. Some justification of the assumption that overlaps in diel activity between species is directly related to predator.prey interactions is also necessary.

The methods require better justification and more detail, in particular on camera spacing, length of deployment, potential loss of independence between data from different cameras, and why more robust analyses weren't applied, such as occupancy modelling.
Density is presented on your figures but it is not clear how this was estimated, and is clearly not true density.

The conclusions and assumptions in the Discussion should be better supported by reference to other studies. There should be more discussion around predator/prey relationships.

Best regards,
Yolanda

My specific comments are:

Throughout the manuscript it is not clear whether the study is on domestic cats (i.e. pet) or feral cats – both terms are used. This must be clarified.
In general there is insufficient detail in the Methods.
Abstract: explain what “previously selected vertebrates” are.
Provide Latin names for species.
Explain what “a 17% shift in activity levels” is.
Introduction
Line 52: delete “recently”
Line 69: delete “and severe”
Lines 70 – 79: the message/point of this paragraph is not clear.
Line 81: change to – “In Australian, 400 vertebrat species were recorded as targets of predation…”
Lines 82-3: Change to “”in the Philippines, members of the orders Chiroptera and Rodentia are the most numerous mammalian species (Heaney et al. 2010). A wide range…..”
Line 87: delete ‘the”
Line 96: Clarify “represent noticeably poorer fauna than that found”.
Line 101 – provide common names.
Line 105: insert “a” before “transition”.
Line 110: delete “using the R package overlap”.
Lines 110-111: this aim is too vague: be more specific and provide some rationale/justification for wanting to look at overlap.
Line 128: substitute ”measuring: with “when”: re-write sentence as: “the driest month is April when approximately 40 mm of rain falls.
Line 144: how was the different in performance between cameras tested?
Line 148: how far apart were cameras in clusters? Cats have large home ranges, so were you counting the same individuals at different cameras?
How long were the cameras deployed and at what times of year?
Was bait used?
Density is presented on figures: how was this calculated? It is certainly not a true density.
Was the accumulation curve calculated on the pooled groups of taxa?
Lines 172-3: If cameras were close would there not be multiple captures of the same individuals across cameras and throughout the year? How has this been accounted for in the calculation of the relative abundance index? Species with larger home ranges, or species that are more active would have relative activity indices biased up.
Explain in this section what NEV and NID is.
Line 179: here it says that species were identified to species, but previously it said this was not possible and they were grouped into taxa. This point requires clarification.
Line 186: explain what the “circular data” are.
Line 189: “pairs” of what?
Line 196: delete “Finally”
Line 218: delete ‘level” and insert “activity in” between “whereas” and “the”.
Line 224: Clarify what a 17% shift in temporal occurrence means.
Line 235: insert “recorded as occurring” between “species” and “on”.
Line 236: insert a comma after )
Line 238: stipulate that these are also camera captures.
Lines 245-250: It is odd the cats are not in the forest. Other studies have shown feral cats occur in forest – these should be mentioned. It is very unlikely that cats would be deterred by steep terrain. Other studies also report on how far domestic cats roam from their houses: these could also be discussed.
Line 252: “weak support as an explanatory variable”. This implies a statistical analysis but this was not done.
Line 255” replace “performing nocturnal activity’ with ‘and are active at night”.
Line 257” avoidance of cats, not avoidance with cats.
Lines 255-257: I don’t find these conclusions entirely logical.
Line 264: Tews et al.’s hypothesis has nothing to do with cats, as implied here.
Lines 269-271: Not a convincing explanation for the absence of cats in forest. Maybe the cameras just didn’t capture them for some reason.
Line 277: what explanation?
Line 290: delete “the” from before “feral”.
The assumption is that the presence of cats is a causative factor, influencing the activity of other animals, but could it also be that both taxa are responding to other factors influencing resource availability that shift depending on habitat

·

Basic reporting

Domestic house cats are important introduced predators in many parts of the world, and this study extends our understanding of their potential impact on the small vertebrates of Bohol Island, Philippines, by using cameras to quantify overlaps in the temporal activity of cats with other species on the island. The basic reporting of the research is sound, but I think could be improved in several (mostly minor) ways, as follows:

Title: Please add 'Philippines' at the end of the title so that readers know straight away where Bohol Island is.

Abstract, line 36: I'm not sure what 'selected' means here. Perhaps 'recorded' would be more accurate.

Abstract, lines 37-38: The scientific name of the barred rail should be given here. Also, as this was the most frequently recorded species, this result should be mentioned in the Results section too.

Abstract, line 39: You need to say here what is meant by 'overlap'. It is clear later that this refers to overlap in diel activity, but this needs to be clarified when first mentioned (and where the Δ symbol is shown).

Abstract, line 43: Rather than 'activate', I suggest replacing with: 'become active'.

Lines 51-52: Replace 'rate' with something like 'proportion'. 'Rate' implies a temporal aspect that is not correct here. The word 'Recently' could also be omitted.

Lines 80-87: This paragraph makes assumptions about the size range of prey that is hunted by cats and, while I do not disagree with the assumptions that are made, a reference or two should be given to provide support. One useful citation may be: Pearre and Maass (1998) "Trends in the prey size-based trophic niches of feral and House Cats Felis catus L." Mammal Review 28: 125-139.

Lines 100-111: This paragraph introduces the ideas that cameras will be used to 'capture' cats and potential prey and co-occurring predator species, and that the camera-data will be used to compare the diel activity of these different groups. This is fine, but it really needs more justification: what can overlaps (or lack of overlaps) in diel activity between species tell us about their likely interactions? What previous studies have used diel activity overlaps between cats, other predators and prey? I am aware of some research in Australia that has explored these questions, and this might help to provide the motivation for the approach used in the present study. See Wang Y and Fisher DO (2012) Wildlife Research 39: 611-620; Brook LA et al. (2012) Journal of Applied Ecology 49: 1278-1286; Greenville AC et al. (2014) Oecologia 175: 1349-1358. There may be more studies of diel overlap involving feral cats; these are studies that I know of.

In the same paragraph, it may also be worth noting why the authors did not use analyses of their camera data that are potentially more powerful than describing simple overlaps in activity. For example, 2-species occupancy models have been used to describe patterns of avoidance among feral cats, their prey and potential competitors (see Lazenby BT and Dickman CR (2013) PLoS ONE 8(4): e59846). It may be that the cameras in the present study were too close together to provide the independent data needed for such analyses, but again I think it would help the authors to build the case for their approach by saying whether other methods had been considered.

Line 140: In the paragraph on sampling procedures, please provide some more information on camera spacing. Individual cameras look to be quite close in Fig. 1; was a minimum spacing considered? Also, how long were individual cameras set to record for, and did individual cameras record for different lengths of time?

In the Results section and in the captions of Figs 4-7, please explain what is meant by 'Density' on the y-axis of each figure. It is clearly not animal density and, from the description at lines 173-176, it is probably not the relative abundance index. Is 'Density' some other index of the numbers of animals that were photographed? In the same figures some explanation is needed of what the shaded areas represent.

In the Discussion, where you note that the results uncover the first tempo-spatial co-occurrences between domestic cats and their potential prey in the Philippines (lines 291-293), it would help to refer to other works (such as those noted above) that have documented such co-occurrences. The results on Bohol Island are certainly of interest, but they are not without precedent elsewhere.

Experimental design

The sampling and experimental design appear to be fine. My only question is (as above) how far apart the cameras were set so that readers can judge how independent they are likely to be.

Validity of the findings

The results are simply described and the conclusions drawn from them are quite reasonable and logical.

Additional comments

This is an interesting study, simply reported. The methods used are fine, but my main suggestion is to better justify why they were used (by reference to previous work) and whether alternative methods that could have further explored patterns in the data had been considered.

·

Basic reporting

The writing is somewhat clunky at times, with sentences that are overly long and complex. I would suggest more editing to make the writing flow more clearly.

I think the introduction needs to focus more on the objectives and why camera traps are being used. Too much is simply just listing the fauna of the Philippines. Also, the objectives seem not to be related - the title is about cats, but then the first objective is about just seeing which species are captured by camera traps - which seems like an unrelated objective.

The authors should explain why they are using camera traps, and what they want to get out of activity levels. For example, what do they predict about the activities of cats, their competitors and their prey? in different zones and in relation to each other.

Experimental design

I have a few questions about the experimental design. The use of zones and categories of land use (agriculture land, primary forest, etc) is confusing. Sometimes the results talk about zones, but sometimes it talks about rural landscape. It would be easier if you just stuck to either Zones or landscapes, because now I'm left wondering what whether the rural landscape you write about spans multiple Zones or just one Zone. If it's only in one Zone - why not just refer to the zone?

Secondly, I'd like more details about where the cameras are located. In the map, they look very close together, which makes me question the independence of the photos they capture. You also need to explain the species accumulation methods more - they are too brief.

Validity of the findings

The finding on the usefulness of camera traps is fine, but again, not really related to the title of this paper (same with the appearance of the pig - which should be a natural history note).

I think more thought needs to go into the discussion - again it would help if you clarified the zones versus land-use categories. I'd like to know what you think is going on with the predator and prey species, and with competitors of cats. If you don't consider civets competitors, why include them in the analysis in the first place?

More thought needs to go into predator prey relationships - is there evidence that cats are negatively affecting native species? Do these species have defenses because they coevolved with civets? Are cats sticking around in one zone because they don't want to climb steep hills or because their prey don't like to cross steep hills into zone 1?Why are there so many more birds at sites with cats than without? Are the cats just going there because the birds are there?

Additional comments

this is a really interesting issue and I think the results should be published. However, this manuscript can benefit from some better organization first to help the reader interpret the results better.
Specific comments:
Style – hard to tell separate paragraphs – can you indent or double space between paragraphs?
Lines 52-54 – suggest rewrite as “The terrestrial vertebrate taxa, which primarily encompasses small to medium sized species, inhabit more than 7100 islands;”
Lines 55-56 – “These species include 213 mammals, 674 birds, etc.
Line 58 – change to “such as Rattus spp. And the domestic cat”
Line 70 – Remove “As described” and move reference to end of sentence.
Lines 72-74 – please relate this sentence more closely to cats specifically
Line 81 – rephrase as “Australia alone has recorded 400 prey species consumed by cats” – also how does this inform your study in the Philippines?
Line 82- “In the Philippines, the mammalian fauna is dominated by members… “
Line 86- remove “face a” and “of”
Lines 80-92 – is there a way to generalize this paragraph better? Instead of just listing different species that could be cat prey – can you say XX% of the mammalian species fall within cat prey range?
Line 94 – one insectivorous rodent species? Why is it singled out?
Line 96 – is Luzon larger?
Line 97-98 – “with Passeriformes forming the largest sub-group at 83 species.”
Lines 105-107 – would be great if your map showed the zones.
Lines 107-111 – You could list your objectives – “Our objectives were to : 1. Create a general inventory ... and Compare the species accumulation curve documented by camera traps to the number of known species previously detected by camera traps, and 3… Also, you can talk about the R package in methods, not intro.
Lines 149-153 – was there only one camera per location? How far apart were the cameras from each other? Could you have two cameras along the same trail? This is important to know to determine independence.
Lines 159-161 – the sentence is a bit confusing because you start out talking about reptiles but then you return to mammals. Better to have the mammal part of the sentence be a separate sentence or just say ground-dwelling terrestrial mammals (that already eliminates bats and arboreal mammals).
Lines 167-169 – do all the rodents have similar activity – probably should show that their overlap is very high – 0.9 or so.
Lines 181-182- what do ACE and ICE stand for? Why do you need 3 estimates?
Line 196 – delete “finally”
Line 197 – that’s cool you found the pig – maybe this should go into a separate paper – a biological note?
Line 201 – you have zones and different types of habitats – that gets confusing to remember. Why not stick with one?
Line 202- transition zone = Zone 2?
Line 224 – I don’t think you can say 17% shift because overlap is not an exact calculation of activity – just a fitted curve. You can say exhibited less activity at night and more during the morning hours like you do in lines 218-219.
Line 229 – “cats exhibit roughly consistent activity…”
Line 234 – in your results, you say you captured 100% of expected mammals, but here you say you didn’t capture the pygmy squirrel. What’s the difference?
Line 237 – To clarify, I might add “compared to known species, camera traps captured 81.8% of ground-dwelling mammalian, avian, and reptilian species, similar to the 86 % captured in the Amazon… etc.” However, I do think it’s a bit unfair to say that since you excluded 3 of 4 reptiles from consideration without explanation.
Line 247-249 – this sentence is very awkward – please rephrase.
Lines 254-257- I’m not sure I understand your justification for why you can’t consider civets competitors to cats – because they’re omnivores and nocturnal? Also you are or are not omnivorous - you can’t be partly an omnivore. Also are you saying that because dogs are with people or omnivores, they also don’t count as apex predators?
Lines 262-264- you’re referring to table 2? I don’t see the birds as being much higher activity wise…Also, your data seems to show there’s plenty of bird and rodent activity where cats are – so maybe they are not too impacted by cats? Do civets not hunt rodents and birds too? Maybe they already have good defenses against a small ground predator.
Lines 277-278 – what explanation do you refer to? Also you can’t say anything about the number of prey species – only the activity index.
Lines 284-285 – Is this with or without cats?
Line 287- it might be more accurate to say the presence of predators resulted in rodents shifting to more nocturnal activities – if cats are more diurnal.
Lines 293 – “we confirmed that camera traps are capable of capturing small bodied…”
Line 297- I would suggest DNA analysis of scat as more detailed and accurate identification of species eaten.
Lines 300-301 – you need to discuss this more. How long have cats been in the islands – are they people’s pets or just feral?
Figure 1 – indicate your zones on this map
Figure 2 – fun picture – identify the species in the caption. However this is not really relevant to the paper.

---

## Round 0.2 · Minor Revisions

Dear Tomáš,

Both of the reviewers have made very helpful and pertinent comments, and all of them should be addressed. I have also edited your manuscript to improve the English and the content, and am attaching it as a pdf. You'll see there are some comments - they overlap with those of the reviewers, and certainly need to be addressed.

You need to explain in the introduction what the spatiotemporal activity patterns will reveal about cats and their competitors and prey. Right now, the set up is that cats eat a lot of native species, and that there will be a comparison of their activities. But why use camera traps and why use activity levels?

I and both reviewers feel that it is necessary to report how far apart the cameras were placed in the paper. Your response letter indicates some cameras are within 20-50 m of each other. It is likely that two cameras that close to each other would often capture the same animal twice. So, if an animal walked in front of one camera and the same species of animal walks In front of another camera a few minutes later, are those two events or one? It is therefore questionable to call events "independent". This limitation should be clearly addressed in the paper, because if several cameras are very close, that could really skew the results. You say that you report number of events in Table 1, but this doesn't appear to be the case, and it is still not clear what an event is.

When you submit your response, rather than just writing "done", indicate what you have done by cutting and pasting the altered text into your response, and indicating the line numbers, so that I am able to find it easily in the manuscript.
I look forward to hearing from you,
Yolanda

·

Basic reporting

This revised manuscript addresses and corrects most of the concerns that I had with the original manuscript, and now provides clearer insight into the potential impact of feral cats on the small vertebrates of Bohol Island, Philippines. The use of camera traps nicely shows overlaps in the temporal activity of cats with other species on the island and hints at the interactions that may be occurring. As before, I think the basic reporting of the research is sound, but some minor issues and grammatical errors still remain; attention to these would help to further improve the manuscript.

Line 55: 'encompasses' should be replaced by 'encompass'. ('Taxa' is plural).

Line 96: I think 'subordinate' is meant here, not 'superordinate'.

Lines 91 and 159: Varanus cumingi is given two common names here, Cuming's water monitor and the yellow-headed water monitor. Please use one name for consistency.

Line 174: Refer here to Table 3, not Table 1.

Line 178: I think 'expected' is meant here, not 'selected'.

Methods section: In describing the use of the 'overlap' program in the last paragraph of the Methods, I am still unclear about how 'Density' is calculated and used as the y-axis on Figs 4-7. The estimation of 'Density' is actually expressed more clearly, to me, in the rebuttal letter than in the text of the manuscript, and I suggest incorporating this slightly longer explanation into the text to help clarify this point for readers.

Table 1: Apart from what seems to be an incorrect reference to Table 1 at line 174, I can't see where Table 1 has been referred to in the text. It should be mentioned in the appropriate place.

Lines 224-225: Please check what is said here. Table 4 and Figure 5 both suggest that the highest overlap zone between cats and ground-dwelling birds is Zone II, not Zone III.

Lines 256-257: This sentence is unclear and needs to be reworded.

Table 2: The table heading says that "Taxa denoted with ‘*’ are those expected for the species accumulation curve." However, no taxa listed in the table have an asterisk; these should be added.

Table 3: Please write 'ground-dwelling birds' rather than 'g.-d. birds'.

Experimental design

As before, the sampling and experimental design appear to be fine. My only question previously, about how far apart the cameras were set, has been addressed in the rebuttal letter. I suggest that this information - cameras were set 20 m to 500 m apart - be added to the text of the manuscript.

Validity of the findings

The results are simply described and the conclusions drawn from them are quite reasonable.

Additional comments

This is an interesting study, simply reported. The authors have done a good job in addressing a number of comments that I raised on the original manuscript, and I thank the authors for their efforts in doing this.

·

Basic reporting

Introduction – I still think the introduction needs to explain what the spatiotemporal activity patterns will reveal about cats and their competitors and prey. Right now, the set up is that cats eat a lot of native species, and that there will be a comparison of their activities. But why use camera traps and why use activity levels – what do the authors expect to learn from these data points.
Line 37 – change to “frequently detected” or “common”
Line 40 – “Cat activity”
Line 45 – no temporal response by cats?
Line 88 – “dogs”
Lines 98-100 – “As an invasive mesopredator, feral cats may be suppressed by apex predators in invaded habitats, which has important wildlife conservation implications, especially in areas where the dominant competitor is declining.”
Line 102 – “and to confirm the presence of cats”

Experimental design

Methods – The authors still did not give information about how far apart the cameras were placed in the paper. But in their response letter, it seems some cameras are within 20-50 m of each other. It seems like two cameras that close to each other would often capture the same animal twice. So, if an animal walked in front of one camera and the same species of animal walks In front of another camera a few minutes later, are those two events or one? It seems like this limitation should be clearly addressed in the paper, because if several cameras are very close, that could really skew the results. You say that you report number of events in Table 1, but i don't see that language and i'm still not clear what an event is (see scenario above). The zones seem to be more clear now.
Lines 168-169 – does it makes sense to group mice, rats and squirrels? Did you look at their activities to see if they had strong overlap? Squirrels seem more diurnal to me, but maybe that’s not the case for these squirrels.

Validity of the findings

Is there any evidence that cats are killed by dogs or civets? Or that civets are superior competitors to cats – should have a reference for that if there is. Otherwise, I think the logical conclusion may be that they do not impact each other. Also, a lot of the prey species are not native – that should be accounted for in your discussion. Remember to always qualify your discussion by stating when you are speculating and by discussion relationships as correlations rather than causations.
Line 210- change previously specified to “expected”
Line 249 – “cats’ preferred habitats”
Lines 260-261- “the absence of cats”
Lines 260-270- it seems like civets are just naturally nocturnal – or nocturnal to avoid humans. Civet activity may have no impact on cats at all, and vice versa. I’m not sure if that’s what you mean by neutral co-occurrence – that neither species impacts the other.
Line 266 – “which suggests their neutral…”
Lines 286-288 – maybe cats were less active during the day to avoid dogs? It would be useful to show how human and dog activity interacts with cat activity.
Lines 293-294 – do you mean that there were a lot more birds in one zone and not many bird detections in the other zone so the activity predictions for one is more accurate?
Line 303 – “spatio-temporal”
Line 309 – “recommended”

Additional comments

I think there are some improvements to this paper since the last version. I think it's helpful to think about why you are using this spatiotemporal analysis. It seems like it is just a tool used, but there's no real set up explaining what you hope to get out of it. There are lots of possibilities to examine - how cats relate to humans, how cats relate to their prey (and vice versa) and how cats relate to their competitors. But the results are not easy to interpret (or they can be interpreted in multiple, and sometimes opposite ways). That's why i think it's important to understand why your'e doing this analysis, what you are expecting it to tell you so you can present a more clear interpretation of the findings.

---

## Round 0.3 · Minor Revisions

Could you please read the attached pdf and incorporate the edits I have made on the sections highlighted in green, to improve the English. You should check that the original meaning is still apparent. Could you also address the inserted comment, where some text requires clarification.

After these minor changes we will be able to Accept the article

---

## Round 0.4 · accepted · Accept

Thank you for making those final edits.The copy editor may also make some suggestions for improving the English, but the messages throughout are now clear. I look forward to seeing it in print.